# CVR-MRICloud: An online processing tool for CO2-inhalation and resting-state cerebrovascular reactivity (CVR) MRI data

Peiying Liu [1,2]*, Zachary Baker[2,3], Yue Li[4], Yang Li[2], Jiadi Xu[2,5], Denise C. Park[6], Babu G. Welch[7,8], Marco Pinho[8], Jay J. Pillai[2,9], Argye E. Hillis[10], Susumu Mori[2,5], Hanzhang Lu[2,3,5]

1 Department of Diagnostic Radiology & Nuclear Medicine, University of Maryland School of Medicine, Baltimore, Maryland, United States of America, 2 Department of Radiology, Johns Hopkins University School of Medicine, Baltimore, Maryland, United States of America, 3 Department of Biomedical Engineering, Johns Hopkins University School of Medicine, Baltimore, Maryland, United States of America, 4 AnatomyWorks, LLC, Baltimore, Maryland, United States of America, 5 F.M. Kirby Center for Functional Brain Imaging, Kennedy Krieger Institute, Baltimore, Maryland, United States of America, 6 Center for Vital Longevity, School of Behavioral and Brain Sciences, University of Texas at Dallas, Dallas, Texas, United States of America, 7 Department of Neurological Surgery, University of Texas Southwestern Medical Center, Dallas, Texas, United States of America, 8 Department of Radiology, University of Texas Southwestern Medical Center, Dallas, Texas, United States of America, 9 Department of Neurosurgery, Johns Hopkins University School of Medicine, Baltimore, Maryland, United States of America, 10 Department of Neurology, Johns Hopkins University School of Medicine, Baltimore, Maryland, United States of America

* peiyingliu@som.umaryland.edu

**Data Availability Statement:** The anonymized data sets necessary to replicate our study findings are shared on Dryad. The data sets can be downloaded at: https://doi.org/10.5061/dryad.wh70rxwqw.

## Abstract

Cerebrovascular Reactivity (CVR) provides an assessment of the brain's vascular reserve and has been postulated to be a sensitive marker in cerebrovascular diseases. MRI-based CVR measurement typically employs alterations in arterial carbon dioxide ($CO_2$) level while continuously acquiring Blood-Oxygenation-Level-Dependent (BOLD) images. $CO_2$-inhalation and resting-state methods are two commonly used approaches for CVR MRI. However, processing of CVR MRI data often requires special expertise and may become an obstacle in broad utilization of this promising technique. The aim of this work was to develop CVR-MRICloud, a cloud-based CVR processing pipeline, to enable automated processing of CVR MRI data. The CVR-MRICloud consists of several major steps including extraction of end-tidal $CO_2$ (EtCO2) curve from raw $CO_2$ recording, alignment of EtCO2 curve with BOLD time course, computation of CVR value on a whole-brain, regional, and voxel-wise basis. The pipeline also includes standard BOLD image processing steps such as motion correction, registration between functional and anatomic images, and transformation of the CVR images to canonical space. This paper describes these algorithms and demonstrates the performance of the CVR-MRICloud in lifespan healthy subjects and patients with clinical conditions such as stroke, brain tumor, and Moyamoya disease. CVR-MRICloud has potential to be used as a data processing tool for a variety of basic science and clinical applications.

**Funding:** This work is supported in part by the National Institutes of Health grants R01 NS115771 (P.L.), R01 AG064792 (H.L.), RF1 AG071515 (H. L.), R01 NS106711 (H.L.), R01 NS106702 (H.L.), UF1 NS100588 (H.L.), RF1 AG006265 (D.P.), R01 DC005375 (A.H.), R01 DC015466 (A.H.), R01 NS084957 (S.M.), P41 EB031771 (H.L.), S10 OD021648 (H.L.). The funders had no role in study design, data collection and analysis, decision to publish, or preparation of the manuscript.

**Competing interests:** The authors have declared that no competing interests exist.

# Introduction

Cerebrovascular reactivity (CVR) is an important index of the brain's vascular health. It measures the ability of cerebral blood vessels to dilate or constrict in response to vasoactive stimuli, which provides complementary information to steady-state vascular parameters, such as cerebral blood flow (CBF) and cerebral blood volume (CBV). In recent years, the measurement of CVR has been increasingly reported in a range of applications, such as arterial stenosis [1–5], stroke [6, 7], small vessel disease [8, 9], brain tumors [10–12], traumatic brain injury [13, 14], substance abuse [15], normal aging [16–18] and dementia [19–22]. More recently, CVR as a candidate biomarker of vascular diseases has also been applied in multi-center settings, such as in the MarkVCID study [23] and the INVESTIGATE-SVDs study [24].

Two of the most commonly used methods for MRI-based CVR mapping are CO2-inhalation CVR (CO2-CVR) and resting-state CVR (RS-CVR) [25]. In CO2-CVR, the subject inhales a mild hypercapnic gas mixture (e.g., 5% CO2, 21% O2, and 74% N2) as an explicit stimulus, while Blood-Oxygenation-Level-Dependent (BOLD) MRI images are continuously acquired [25]. Since CO2 is a potent vasodilator, the BOLD signal is expected to increase with CO2 inhalation and, by quantifying the extent of BOLD signal change, a measure of CVR can be obtained. RS-CVR is an emerging CVR method that exploits spontaneous fluctuations in breathing rate and depth, which alters arterial CO2, and determines the corresponding BOLD signal change in the resting-state BOLD time series [26, 27]. While CO2-CVR has the advantage of a well-defined stimulus paradigm and more robust signal, RS-CVR possesses the benefit of not requiring additional apparatus related to CO2 inhalation inside the MRI scanner. Both CVR methods have unique values in different applications.

To date, CVR data processing is considered a niche skill and most studies of CVR have used in-house scripts or software [28], creating challenges for reproducible and repeatable research and for researchers and clinicians with less skills in software programming. Therefore, a processing pipeline that is automatic and provides a wide spectrum of quantitative image and regional outputs is highly valuable and can potentially facilitate broader applications of this promising technique. Cloud-based computation represents a new model of neuroimage processing [29]. Compared to the traditional model of downloadable software package and local computer processing, the cloud-based platform utilizes software, computational, and storage resources on the remote computer for data processing and therefore mitigates the requirements on the user computer [30]. The user inputs are typically provided by uploading raw images to the cloud server as well as entering key acquisition parameters on a web interface. Similarly, the user can access the outputs of the processing by downloading text, table, or image files from the cloud website. Previous work has developed several cloud-based tools for the processing of T1 segmentation, diffusion, and arterial spin labeling (ASL) MRI, demonstrating the feasibility and applicability of cloud computing in the context of medical image processing.

In this work, we developed a cloud-based, publicly accessible tool for the processing of CO2-inhalation and resting-state CVR MRI data, referred to as CVR-MRICloud. Here we describe the framework of the pipeline, processing algorithms, and its outcomes, and demonstrate the performance of the tool on representative data of brain aging and cerebrovascular diseases.

# Methods

## General descriptions of CVR-MRICloud

An implementation of this pipeline can be found at https://braingps.mricloud.org/cvr.v5, with the computer architecture illustrated in Fig 1. A front-end web server is responsible for

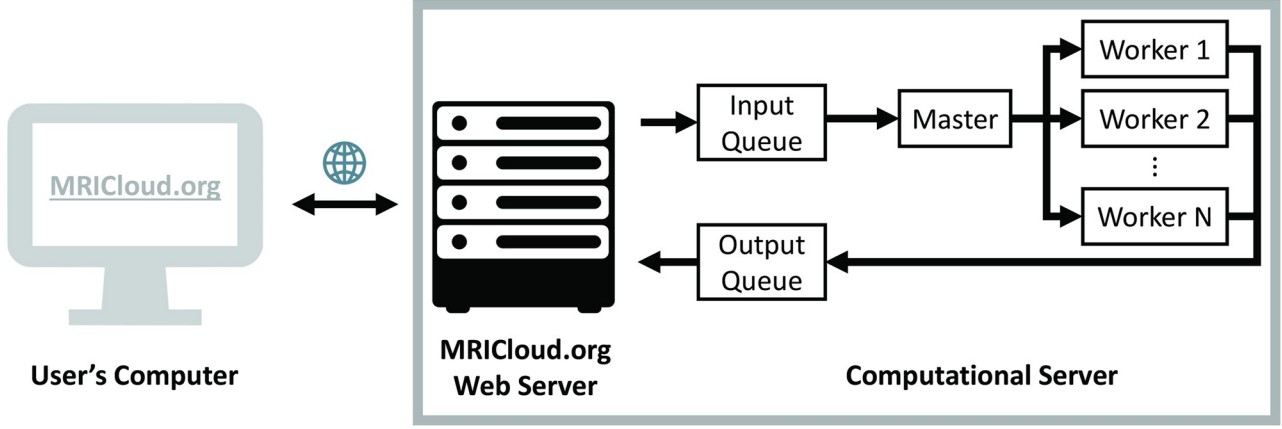

**Fig 1. Illustration of the general architecture of the CVR-MRICloud platform.**

providing functionalities to allow interactions with a user, such as file upload, version selection, and entry of image acquisition parameters. The required image file format is ANALYZE (.hdr and.img) which does not contain protected health information (PHI) and thus, is compatible with Health Insurance Portability and Accountability Act (HIPAA) regulations. The input data are processed by a separate computational server, which then transfers outcome parameters to the front-end server for the user to download. The users can only access the outcome parameters from the data they had uploaded. The data and outcome parameters will be automatically deleted after 60 days from the upload if the users choose not to delete them manually.

The deployment of CVR-MRICloud is conducted in a version-controlled manner. When a new version is released, it will have a specific version number and all older versions are still available on the CVR-MRICloud website.

The CO2-CVR and RS-CVR processing pipelines used an identical computational architecture, but the detailed algorithms have some differences. Below we separately described these algorithms.

## Procedure of CO2-CVR data processing

A flowchart of the CO2-CVR processing pipeline is illustrated in Fig 2. The pipeline contains three major components, including automatic analysis of CO2 recording to obtain EtCO2

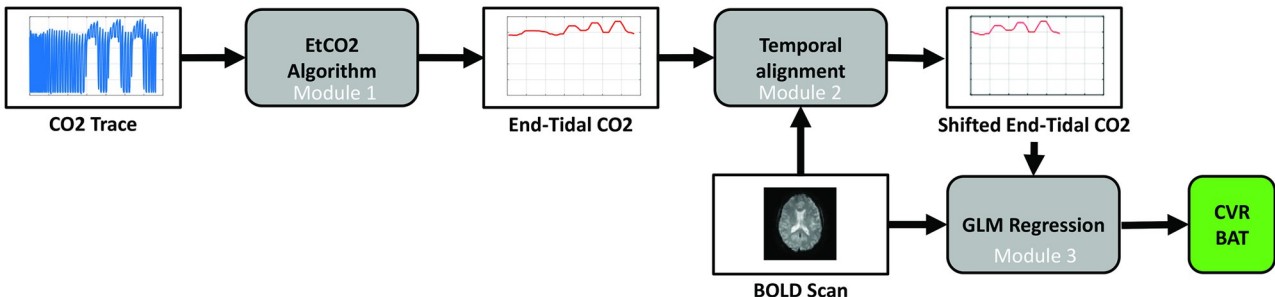

**Fig 2. Flowchart of the data processing pipeline for CO2-CVR MRI data.** Details of the algorithms in Module 1 through 3 are explained in the main text. CO2 –carbron dioxide; EtCO2 –end-tidal CO2; BOLD–Blood Oxygenation Level Dependent; GLM–general linear model; CVR–cerebrovascular reactivity; BAT–bolus arrival time.

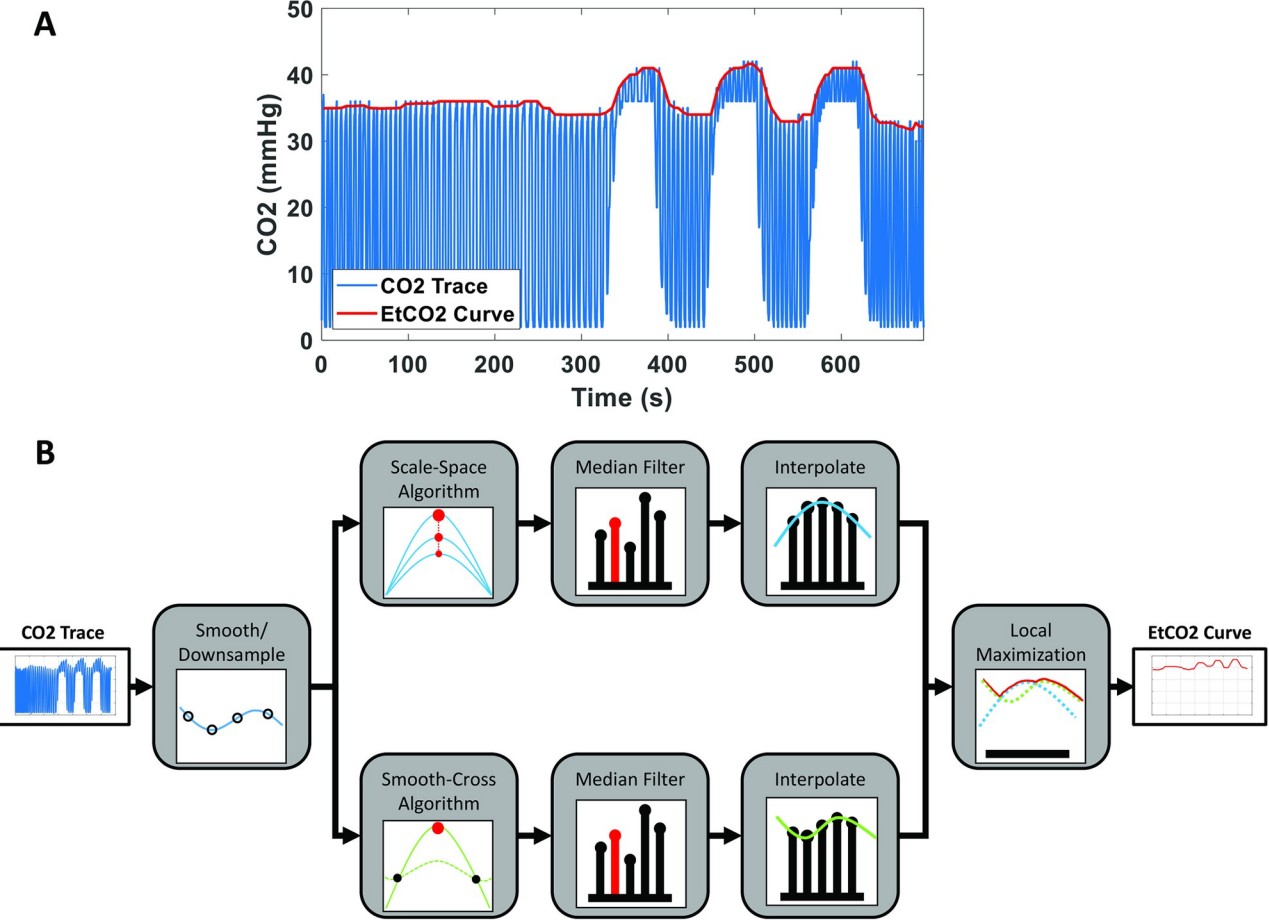

**Fig 3. Extraction of EtCO2 time course from CO2 recording.** (A) Example of raw CO2 recording (blue curve) and the extracted EtCO2 time course (red curve). (B) Flowchart of the EtCO2 extraction processing.

time course (labeled as module 1 in Fig 2), temporal alignment between BOLD and EtCO2 time courses (module 2), and computation of CVR using BOLD and EtCO2 data (module 3).

**Extraction of end-tidal CO2 (EtCO2) time course from CO2 recording.** Fig 3A illustrates an example of CO2 data used in the CVR data processing. The blue, high-frequency curve represents the raw CO2 recording. This recording is typically obtained by placing a small sampling tubing near the mouth or nose of the subject during the experiment, while a CO2 monitor located outside the MRI room samples the air with a pump and analyzes for CO2 content. The sampling frequency is dependent on the specific CO2 monitor used in the experiments, but it generally ranges from 10 to 200 Hz. Each cycle in the curve represents one breathing cycle, with CO2 content of the inhaled air corresponding to the troughs and that of exhaled air corresponding to the peaks. The period of curve is dependent on the breathing rate and typically ranges from 3 to 8 seconds.

The red curve in Fig 3A is referred to as end-tidal CO2 (EtCO2) time course. It reflects the amount of CO2 in the lung, which approximates the CO2 content in the arterial blood. EtCO2 is typically obtained from the raw CO2 recording by extracting peaks, i.e. CO2 in the exhaled air, on the curve.

Fig 3B shows a flowchart of methods to extract EtCO2 from the CO2 recording. We first smoothed (using 100ms-moving window) and resampled the CO2 recording into a pre-set

frequency of 10Hz. Given that typical breathing frequency is <0.3Hz, this sampling rate is expected to be sufficient for the computation of $CO_2$ peaks in the breathing cycle. An additional benefit of the resampling is that the signal-to-noise ratio (SNR) of the $CO_2$ time course is improved.

Next, two separate algorithms were used to identify the peaks in the $CO_2$ time course. We found that each algorithm has advantages in detecting peaks under certain signal patterns, thus by using both algorithms and selecting the higher value between the two outcomes the robustness of the EtCO2 computation pipeline can be enhanced. Algorithm I, referred to as the Scale-Space algorithm [31], uses a single criterion: $v(n) = max[v(n-1) \, v(n) \, v(n+1)]$, to detect all local maxima in the $CO_2$ time course. Then, the time course was smoothed and the local maximum detection algorithm was repeated. If a particular local maximum was still identified after smoothing, its "peak likelihood score" was increased. This process is repeated 30 times and a time course of "peak likelihood score" was obtained. A threshold was applied on the "peak likelihood score" time course to obtain a binary vector of $CO_2$ peaks. The corresponding EtCO2 values at these peaks are also obtained.

Algorithm II, referred to as the Smooth-Cross algorithm, first applies a moving-average filter with window size of 10s. The window size of 10s was chosen as a trade-off between the degree of smoothing and the sensitivity during the room air-$CO_2$ transition periods. The intersection points between the moving-averaged and the original traces then allow the dividing of the $CO_2$ trace into breath-by-breath segments. The peak $CO_2$ value within each segment can then be identified and the corresponding time was labeled to form a binary vector, similar to the outcome of Algorithm I.

Next, the peak $CO_2$ vector obtained by each algorithm was subject to a five-point median filtering so that any peaks as a result of partial breath were removed. Since the intervals of breathing cycle are variable and can be different across individuals, we then interpolated the outputs into a time series of 1Hz. These steps were performed on the outcomes of Algorithms I and II separately.

Finally, the higher value from Algorithms I and II at each point in the interpolated time series was selected to form the EtCO2 curve (Fig 3B), as the EtCO2 curve represent the top envelope of the $CO_2$ trace.

**Temporal alignment between BOLD and EtCO2 time courses.** Before the EtCO2 curve obtained in the previous step can be used for the estimation of CVR, it must be first aligned with the BOLD time course. This is needed because the EtCO2 signal is measured from the lung and the BOLD signal is measured from the brain, and thus a signal shift between them is expected based on physiology. This processing step therefore aims to temporally align the EtCO2 to BOLD time-course. Preprocessing of the BOLD data used SPM12 and included motion correction and spatial smoothing (with 8mm FWHM Gaussian kernel). A whole-brain mask was also obtained by segmenting the mean BOLD image and adding together the gray matter, white matter and CSF masks, which was then applied to the BOLD image series to calculate a whole-brain-averaged BOLD time course.

The alignment of EtCO2 and whole-brain BOLD time courses used the following General Linear Model:

$$BOLD(t) = \beta_1 \overline{\overline{EtCO2(t,s)}} + \beta_2 \ell + \beta_0 \qquad (1)$$

where $\ell$ is a linearly ascending term, i.e. $[-(N-1)/2, -(N+1)/2, \ldots, (N-1)/2]$ and N is the total number of time points. The double bar accent indicates a zero-meaned signal. $s$ is the time shift of the EtCO2 time course and is detailed below.

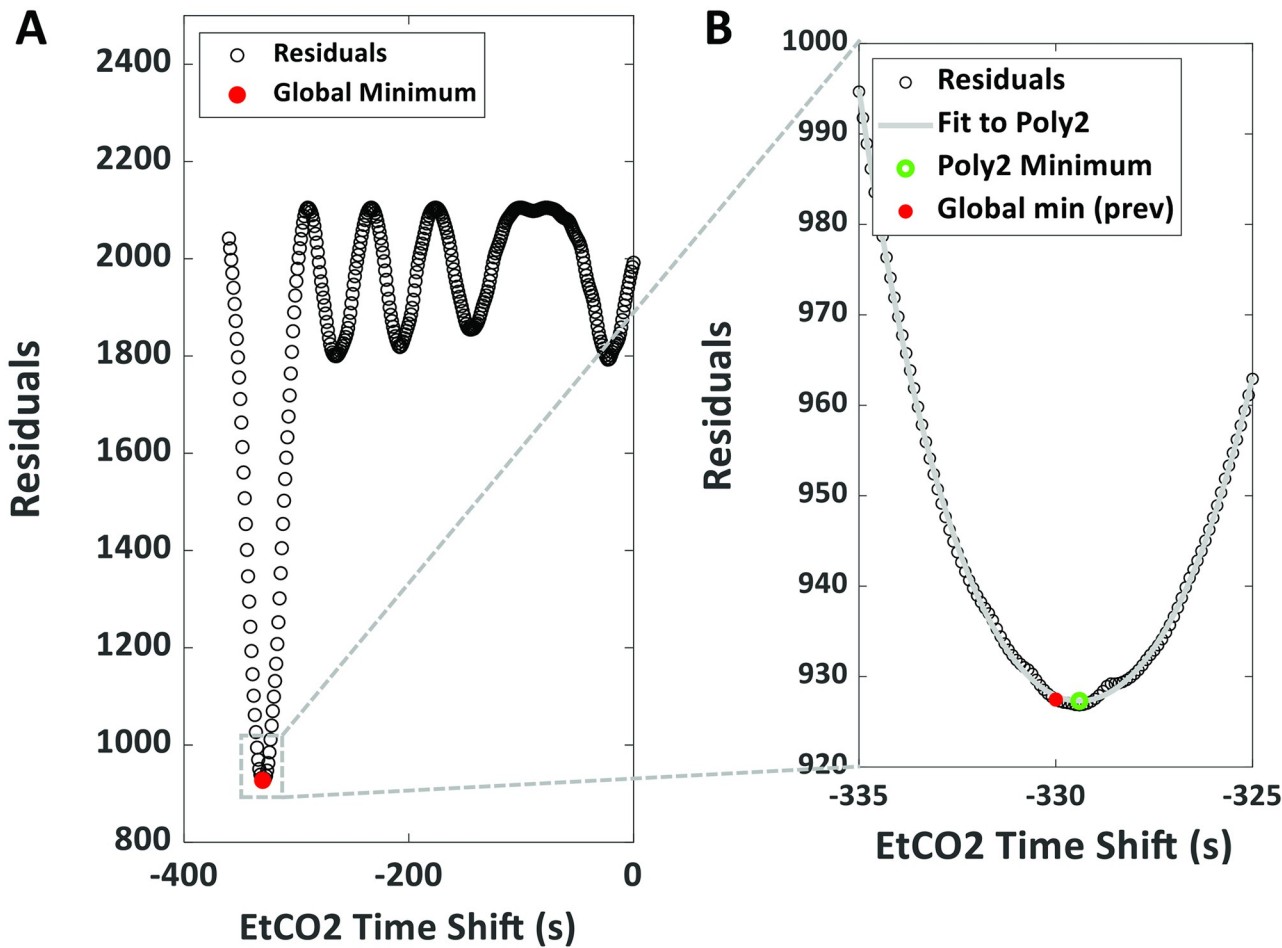

**Fig 4. Example of the two-step processing for the temporal alignment between BOLD and EtCO2 time courses.** (A) Plot of the residual error from the GLM in Eq [1] as a function of time shift in a representative subject. Red dot indicates the time shift that reveals the minimal residual error. (B) Results of the second-step alignment. The plot displays the residual error at each time shift within the 10s range (box in (A)) around the minumal point determined in the first step (red dot). The final optimal time shift (green circle) was determined from the second-degree polynomial fitting of the residuals.

In order to find an appropriate time shift to align these two signals, a two-step process is performed, as illustrated in Fig 4. First, the GLM analysis described in Eq [1] was repeated multiple times, after a step-wise shift (*s*) of the EtCO2 time course relative to BOLD for all possible values (i.e., from 0 to EtCO2 time course duration–BOLD time course duration). The step size was set to be 1 second, in accordance with the sampling rate of the EtCO2 data. After each shift, the EtCO2 time course was resampled (linear interpolation) to the BOLD intervals (i.e. TR), the regression was performed, and the residual error from the fitting was recorded. The shift with the minimal residual error was identified and used as a preliminary "optimal shift value" (Fig 4A). As a second step, a smaller EtCO2 shift step of 100 ms is used and the searching range is set to be +/-5 seconds. The GLM analysis was repeated at each of these finer shift values. The residuals are then fitted to a second-degree polynomial, and the lowest point on the polynomial is used as the final optimal time shift (Fig 4B), similar to the work by Donahue et al. [32]. This two-step procedure is expected to yield more reliable optimal shift value than using the first step alone. EtCO2 time course shifted by this optimal value was then used as the regressor in computation of CVR (see below).

**Computation of CVR.**  Whole-brain CVR is obtained by a GLM between whole-brain averaged BOLD signal and global-shifted EtCO2, using Eq [1] above. The coefficients $\beta_1$ and $\beta_0$ from Eq [1] are then used in the following formula to calculate CVR in the units of %/mmHg:

$$CVR = \frac{\beta_1}{\beta_0 - \beta_1 * [mean(EtCO2) - baseline(EtCO2)]} * 100 \qquad (2)$$

Where $mean(EtCO2)$ and $baseline(EtCO2)$ are the mean and bottom 25% average of the EtCO2 curve, respectively. Here the average of the bottom 25% EtCO2 time points is considered the baseline EtCO2 value, instead of the minimum, to mitigate the influence of unusual breathing pattern in the experiment (for example, a single deep breath at any time during the experiment can result in very low EtCO2 value).

Regional CVR is also calculated. A T1-segmentation MRICloud was used to parcellate the brain into different numbers of segments, ranging from 52 to 289 depending on the level of parcellation [8]. Once the ROI corresponding to each segment is obtained, the BOLD signal was spatially averaged. The resulting time course was aligned to EtCO2 as described above and subject to GLM analysis in Eq [1]. Regional CVR was then calculated with Eq [1]. Note that the EtCO2 curve was aligned to each specific ROI to account for regional variations in the hemodynamic response time, e.g. white matter may have a delayed BOLD response compared to the gray matter [33].

Voxel-wise CVR map was computed using two different methods. One method used the globally-shifted EtCO2 and performed a GLM with voxel-by-voxel BOLD time course. The other method used a voxel-wise-shifted EtCO2 for the GLM analysis [25]. In the voxel-wise-shifted analysis, the global-shifted EtCO2 is used as a starting point and it is then further shifted within a range of -5 to 30s to identify the best shift for each voxel. This search range is relatively narrow to avoid local minimum due to noise and was decided based on previous reports of inter-voxel variations in CO2 response [33]. Note that the voxel-wise-shift analysis also yields a "temporal shift map", which may also contain physiologically useful information similar to bolus arrival time (BAT) map in dynamic-susceptibility-contrast (DSC) and arterial-spin-labeling (ASL) MRI. It should be pointed out that both global and voxel-wise shift approaches have their advantages and disadvantages [25]. The global-shifted EtCO2 is the more robust as it is based on the whole-brain averaged BOLD time course and the SNR of the BOLD data is expected to be excellent. On the other hand, voxel-wise-shifted EtCO2 considers regional/inter-voxel heterogeneity; but the signal can be noisy and thus optimal shift obtained may have poor reliability [25].

Additionally, voxel-wise CVR and temporal shift maps were transformed into the individual's MPRAGE space and the Montreal Neurologic Institute (MNI) standard space. Aside from the absolute value CVR maps, relative CVR maps were also obtained by dividing each voxel by the map's average value.

To provide an index of the overall quality of the CVR data, partial correlation coefficient (CC) between the whole-brain BOLD time course and the global-shifted EtCO2 after factoring out the linear drift, as described in Eq [1], was computed.

## Procedure of RS-CVR data processing

A flowchart of the RS-CVR data is illustrated in Fig 5. Preprocessing of the BOLD data was identical to that described in the CO2-CVR pipeline. However, RS-CVR data did not have associated CO2 recording, thus EtCO2 was not used in the data processing. Instead, the EtCO2 term in Eq [1] was replaced with a surrogate derived from the whole-brain BOLD time

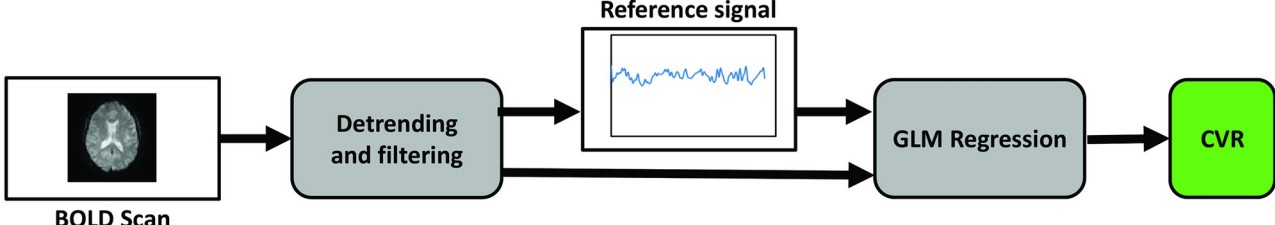

**Fig 5. Flowchart of the data processing pipeline for RS-CVR.**

course. Specifically, the whole-brain BOLD time course was linearly detrended and a low-pass filter with a cutoff frequency of 0.1164Hz is applied. The temporal filtering frequency of [0, 0.1164 Hz] has been found to yield the highest spatial correlation between RS-CVR maps and CO2-CVR maps [34]. Next, the resulting time course was rescaled as $s_0 = (s - mean(s))/(2|s - mean(s)|/\sqrt{N})$, where N is the total number of dynamics, and the operator $||K||$ indicates 2-norm. The purpose of the rescaling is to ensure that the resulting time course has a zero mean and a 2-norm of sqrt(N)/2. This time course is referred to as the reference signal afterwards. A GLM analysis was then performed on each voxel:

$$\frac{\Delta BOLD}{BOLD} = \beta_1 \cdot reference\ signal + \beta_2 \cdot motion + \beta_3 \cdot t + \beta_0, \tag{3}$$

where the voxel-wise BOLD time course is the dependent variable and the reference signal is the independent variable. Six motion vectors (3 displacements and 3 rotations) and a linear trend were also added as covariates. RS-CVR is then obtained as $\frac{\beta_1}{\beta_0} * 100$. The voxel-wise RS-CVR value was further divided by the whole-brain average value to yield relative CVR map.

The relative CVR map was further transformed into the individual's MPRAGE space and the MNI standard space. Regional CVR is also calculated by averaging the relative CVR values within each ROI obtained from the T1-segmentation MRICloud pipeline. Similar to that in the CO2-CVR quantification, regional relative CVR values are calculated for different numbers of segments, ranging from 52 to 289 depending on the level of parcellation.

## Pipeline testing

CVR-MRICloud was tested using data collected in several different studies reported previously [16, 35–37]. For CO2-CVR pipeline testing, 203 datasets from a lifespan cohort (age 50.9±19.8 years, 79 males, 124 females) was used. The CO2-inhalation paradigm used interleaved blocks of 60s room-air breathing and 60s of hypercapnic gas (5% CO2, 21% O2, and 74% N2), for a scan time of 7min. The BOLD MRI sequence used a TR of 2000 ms, a TE of 25 ms, and a voxel size of 3.44×3.44×3.5 mm$^3$, with a duration of 5min performed on a Philips MRI system. In addition, the CVR-MRICloud pipeline was applied to data from a brain tumor patient (58 years old, male) and a patient with Moyamoya disease (55 years old, female). For RS-CVR pipeline testing, resting-state BOLD data from the above-mentioned lifespan cohort, repeated scans from a healthy young subject (28 years old, male), and a patient at 24 weeks after stroke (56 years old, male) were used. These studies have been approved by the institutional review board (IRB) of Johns Hopkins University and University of Texas Southwestern Medical Center. Informed written consent was obtained for each participant.

## Results

### Extraction of EtCO2 time course from raw CO2 recording

Fig 6 illustrates the processing results of EtCO2 in two representative subjects. In the first subject (Fig 6A), the breathing rhythm of the participant was consistent throughout the experiment. Thus, both algorithms used performed reliably. In the second subject (Fig 6B), on the other hand, the individual's breathing rate was more variable. Thus both algorithms generated some incorrect peaks (red arrows). The median-filter processing was able to mitigate the errors to some extent (green arrows), but certain degree of under-estimation in the CO2 peak is still present. The use of the higher value between the two algorithms can further reduce the occurrences of under-estimation. Upon visual inspection of all processed CO2 data described in this

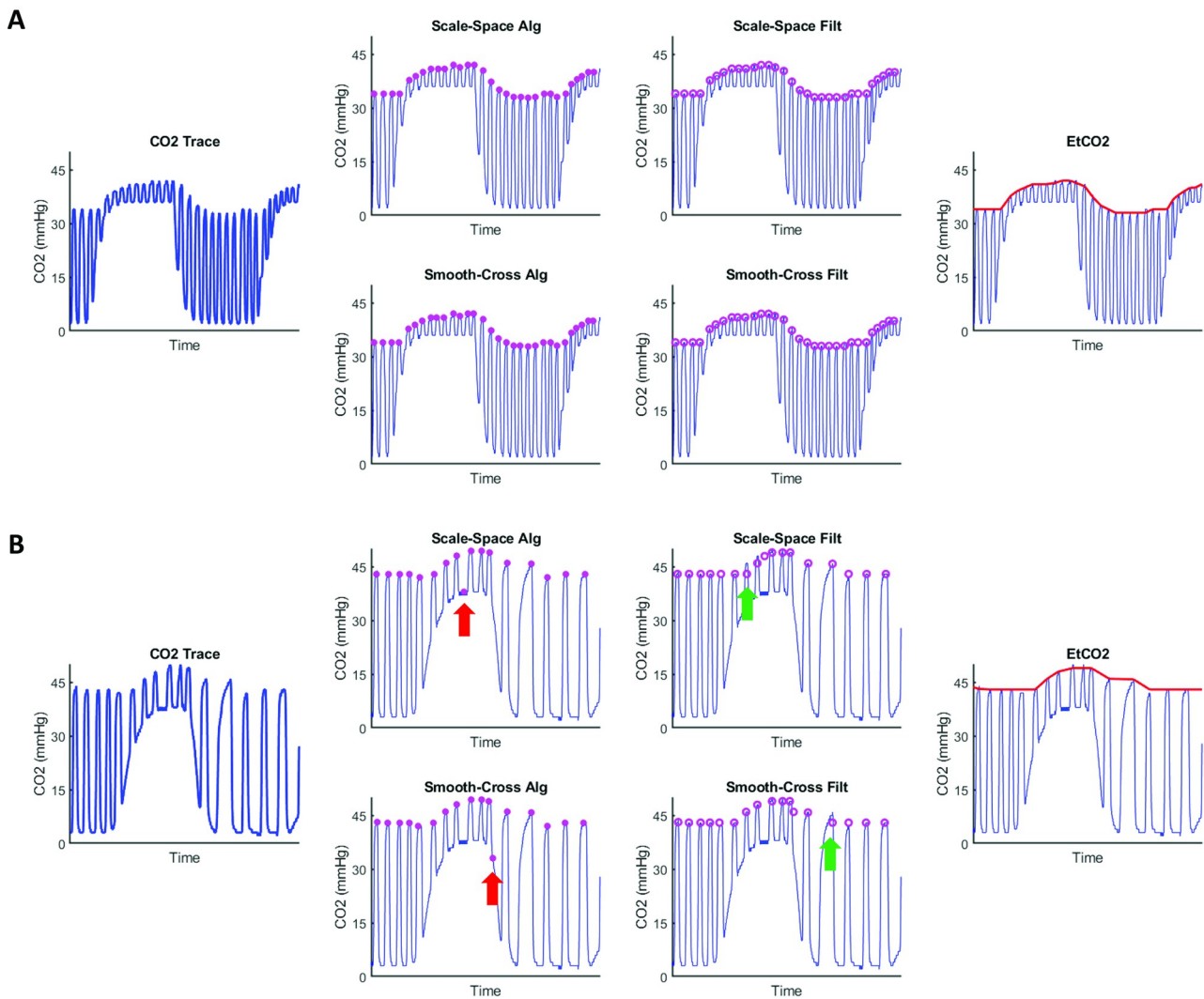

**Fig 6. Two examples illustrating the performance of the EtCO2 extraction algorithm.** (A) Results from a subject with consistent breathing intervals. Both the Scale-Space and Smooth-Cross algorithms detected the peaks correctly. (B) Results from a subject with variable breathing intervals. Both the Scale-Space and Smooth-Cross algorithms showed instances of underestimating the true EtCO2 curve (red and green arrows). These underestimations were corrected by the final step, which selects the higher value from the two outputs. Plots from left to right: CO2 trace, initil peaks identified by the Scale-Space and Smooth-Cross algorithms, peaks after median filtering, outputs after interpolation and finding the higher value between two algorithms (i.e. the EtCO2 curve).

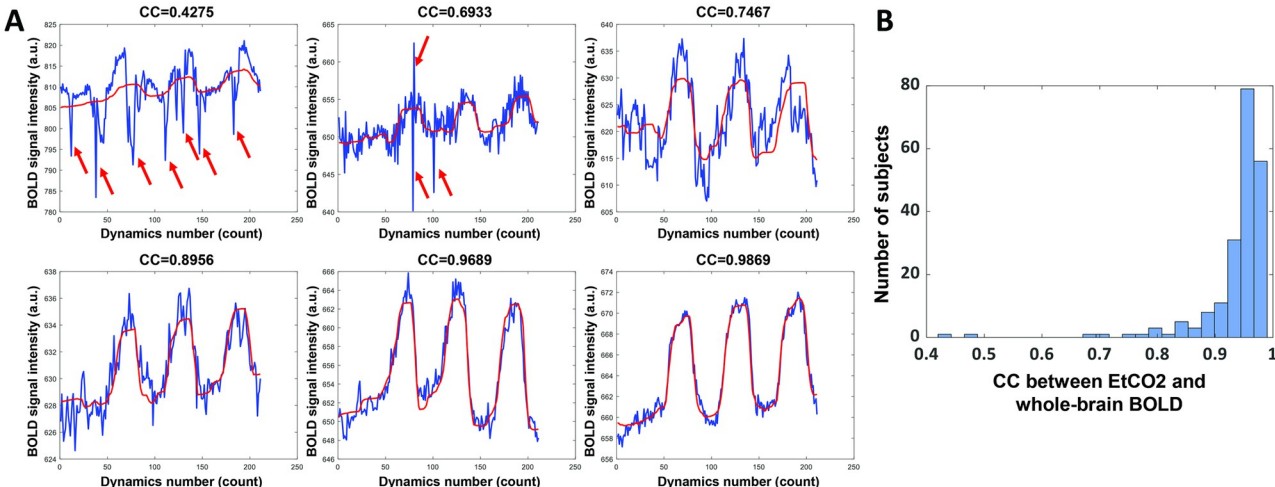

**Fig 7. Performance of the temperal alignment between EtCO2 and BOLD time courses.** (A) Examples of the optimally shifted EtCO2 curves with the corresponding whole-brain BOLD time courses. Six examples with ascending data quality are shown. Red arrows indicate BOLD signal spikes due to subject motion, which result in lower correlation coefficients (CC) between EtCO2 and whole-brain BOLD time courses. (B) Histogram of the CC in 203 healthy subjects.

study (N = 203), we found that the pipeline was successful in finding the CO2 trace's upper envelope, i.e. EtCO2, in all subjects.

## Temporal alignment between EtCO2 and BOLD time courses

Fig 7A shows 6 examples of optimally shifted EtCO2 curves together with the corresponding whole-brain BOLD time courses, stratified by ascending cross-correlation coefficients (CC) between EtCO2 and BOLD time course. For the entire cohort of 203 subjects, the CC were 0.937±0.067 and their histogram is shown in Fig 7B. It can be seen that generally the alignment results were excellent. The cases where low CC values were observed were primarily due to noisy BOLD time course, presumably due to subject motion (e.g. red arrows).

## CO2-CVR

Examples of CO2-CVR maps produced by CVR-MRICloud are displayed in Fig 8A, showing the same six participants illustrated in Fig 7A. These maps have been transformed to MNI space. As can be seen, the quality of the CVR map is directly related to the CC between EtCO2 and whole-brain BOLD time course. Individuals with a higher CC tend to yield CVR maps of better quality. Fig 8B shows time shift maps obtained by voxel-wise shift analysis between the EtCO2 and BOLD time course.

Quantitative values of whole-brain CVR are plotted as a function of age in Fig 9A. The participants were grouped into decades. It can be seen that CVR decreases with age. Fig 9B show decade-by-decade averaged CVR maps.

Fig 10 demonstrates the feasibility of CVR-MRICloud in processing CVR data from pathological conditions. Fig 10A presents a patient diagnosed with a glioblastoma in the left hemisphere. The tumor is clearly seen as a T1-hypointense region on the anatomical MPRAGE image and displays prominent regional CVR reduction relative to surrounding normal tissue on the corresponding CVR map. Fig 10B presents a patient diagnosed with unilateral Moyamoya disease. The angiogram shows an occluded left middle cerebral artery in the left

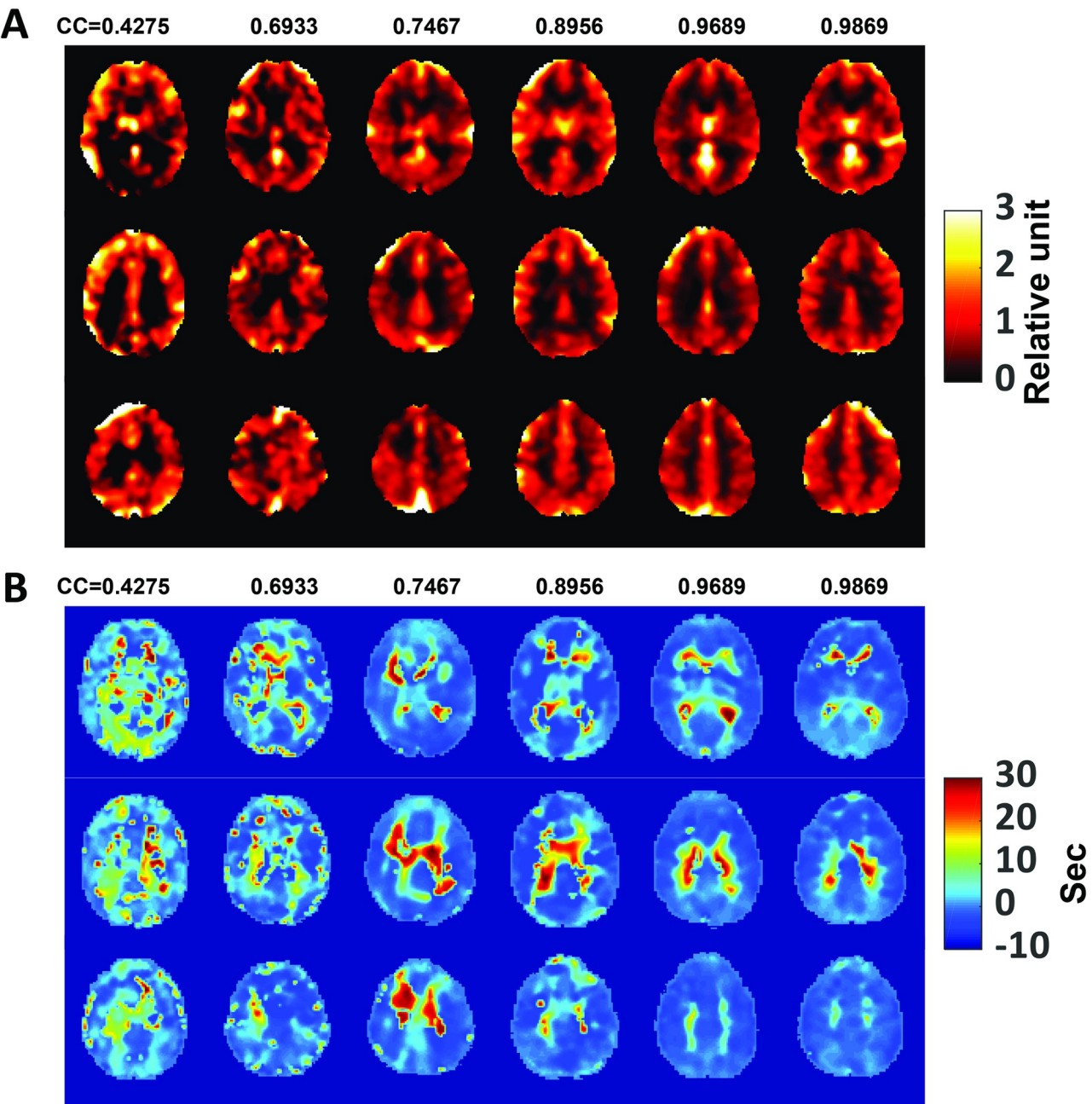

**Fig 8. Examples of CO2-CVR results.** (A) Relatiev CVR maps in MNI space from the same 6 subjects shown in Fig 7. (B) Time-shift maps from the same subjects.

hemisphere. Accordingly, the CVR map demonstrates diminished signal in the left hemisphere, notably in the regions perfused by the left middle cerebral artery.

### RS-CVR maps

Examples of RS-CVR maps produced by CVR-MRICloud are shown in Fig 11. It can be seen in Fig 11A that relative CVR maps can be obtained reliably from resting-state data. Fig 11B

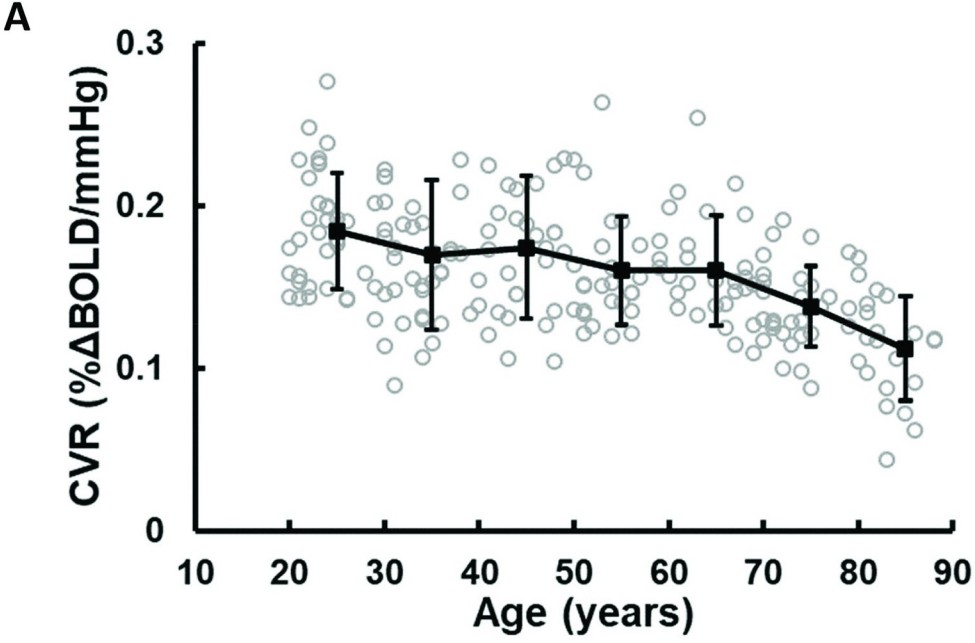

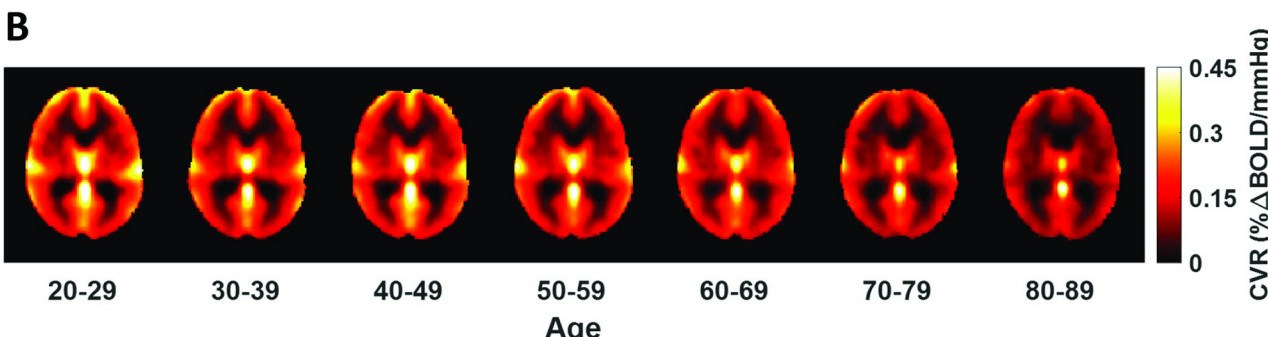

**Fig 9. CO2-CVR results from 203 healthy subejcts.** (A) Decade-by-decade differences in whole-brain CVR values. Mean and standard deviation of each decade are shown in black. Individual CVR value of each subject is shown in gray circle. (B) Decade-by-decade averaged CVR maps. N = 34, 36, 29, 29, 26, 29 and 20, respective, for each decade.

demonstrates good reproducibility of RS-CVR maps acquired from consecutive resting-state BOLD scans in the same healthy subject.

RS-CVR map from a clinical stroke patient is shown in Fig 12. Prominent cortical and subcortical CVR reduction corresponds to the abnormal T2-hyperintense infarcted region in this stroke patient.

## Discussion

In this work we developed CVR-MRICloud, a web-based data processing pipeline for CVR experiments employing either hypercapnic gas inhalation or resting-state MRI. The pipeline encompasses all major steps of CVR processing including CO2 recording analysis, BOLD image pre-processing, temporal alignment between CO2 and BOLD time courses, and computation of whole-brain, regional, and voxel-wise CVR results. These algorithms are fully automated, thus are ideally suited for cloud-based computing. The processing procedure also considers timing differences between different brain regions and tissue types (i.e. gray versus

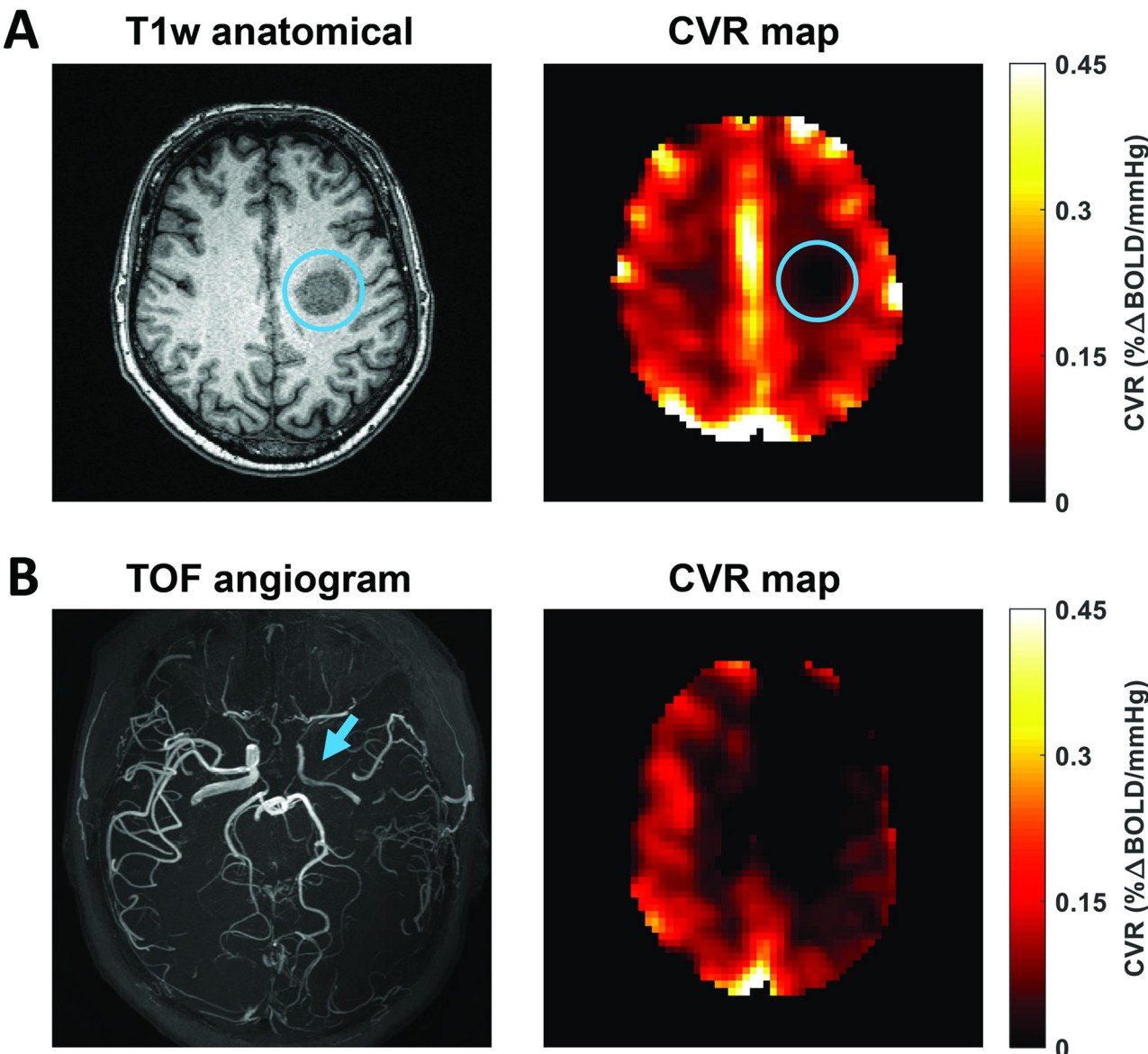

**Fig 10. Examples of CO2-CVR maps in patients with brain disorders.** (A) Anatomical image and CVR map from a patient with glioblastoma (58 years of age, male). Blue circles indicate tumor region. BOLD imaging parameters were: TR/TE = 1550/21ms, resolution = 3.2x3.2x3.5mm$^3$, 362 dynamics. (B) Time-of-flight angiogram and CVR map from a patient with Moyamoya disease (55 years of age, female). Blue arrow indicates right MCA stenosis. BOLD imaging parameters were: TR/TE = 1510/21ms, resolution = 3.2x3.2x4.2mm$^3$, 372 dynamics.

white matter), and provides an estimation of the time shift map, which is thought to be related to bolus arrival time of the brain vasculature and may be physiologically meaningful.

To our knowledge, CVR-MRICloud is the only web-based, fully automated CVR processing tool to date. Many methods of CVR processing, including those from the authors' group [38], contain certain steps of manual processing, e.g. identification of EtCO2 time course from the raw CO2 recording, and often require custom scripts. Other methods such as those in Functional Software Library (FSL) suites [33, 35, 39] requires the installation of software packages on specific operating systems, e.g. Linux. CVR-MRICloud provides an operating-system independent processing platform and can work with any electronic device that has a web-browser

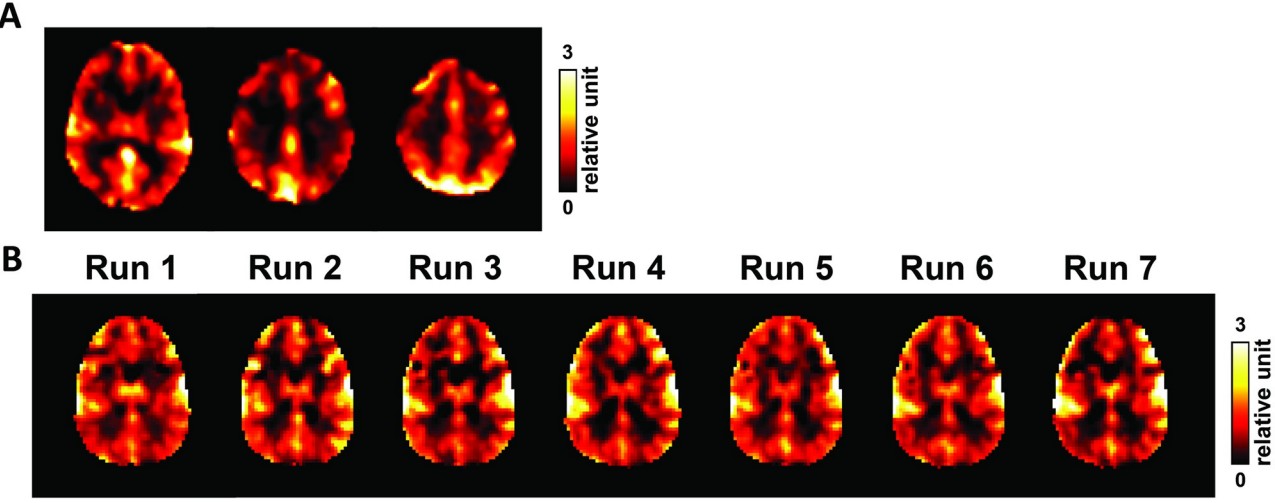

**Fig 11. Examples of resting-state CVR maps.** (A) CVR map from an healthy subject (28 years of age, male). BOLD imaging parameters were: TR/TE = 2000/25ms, resolution = 3.4x3.4x3.5mm$^3$, 154 dynamics. (B) CVR maps from repeated resting-state runs in a healthy subject (28 years of age, male). BOLD imaging parameters were: TR/TE = 1000/25ms, resolution = 3.4x3.4x5mm$^3$, 300 dynamics.

application. Minimal experience is required in order to use this tool. An inter-rater evaluation performed by 4 raters with no previous CVR experience revealed that the ICC of CVR values was >0.9959, suggesting that CVR-MRICloud is user-friendly and the results are not user-dependent [37]. The RS-CVR pipeline has also been utilized by researchers with no previous experience with CVR and yielded multiple publications [40–42]. CVR-MRICloud also presents minimal requirements on the device's CPU, memory, and storage capacity, as all computations are performed on the remote computer provided by the developer. For the CO2-CVR

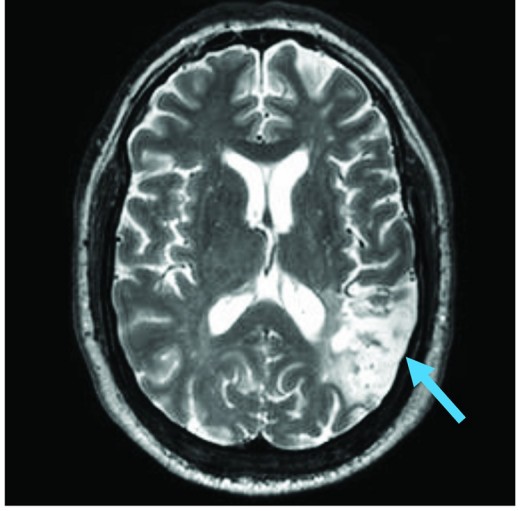

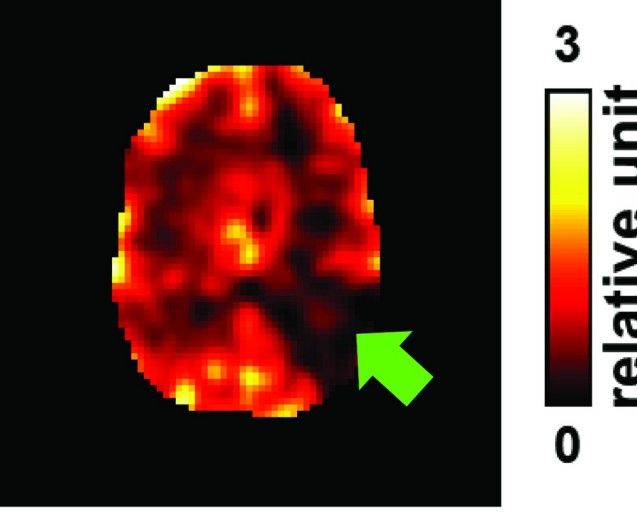

**Fig 12. Resting-state CVR map from a patient with stroke (56 year of age, male).** The MRI was performed at 24 weeks after stroke onset. Infarcts were identified at temporal lobe, insular region and frontal operculum/inferior lobe. Arrows indicate stroke lesion region in temporal lobe. BOLD imaging parameters were: TR/TE = 2000/30ms, resolution = 3x3x4mm$^3$, 210 dynamics.

datasets obtained in the lifespan study with the imaging matrix of 64x64x43 and 211 dynamics, the processing time is about 16min for one dataset. The results of the lifespan study revealed age-related CVR decrease, consistent with previous report [16].

CVR-MRICloud aimed to incorporate the latest advances in CVR processing. The extraction of the EtCO2 time course used two state-of-the-art algorithms, the Scale-Space algorithm [31] and the Smooth-Cross algorithm, which can account for various artifacts in the CO2 recording such as those due to partial breath and irregular breath. The alignment of the BOLD and EtCO2 time courses used a two-step processing and employed the polynomial fitting method similar to that used by Donahue et al. [32], which is potentially more robust to noise than the simple minimum-searching method. The model to compute CVR value is based on linear regression between CO2 (as input) and BOLD signal (as output), but also considered linear drift of the MRI time course that is common in fMRI and CVR data. Motion vectors were not included as covariates in CO2-CVR processing because, in our experience, they can show similar time course as the EtCO2 curve (see S1 Fig for examples) and if included in the GLM analysis, may lead to underestimation of CVR. The CVR-MRICloud pipeline also takes into account previous observations that different regions in the brain, in particular between gray and white matter regions, show a relative temporal shift in their time courses. This temporal shift is well-known from other MRI techniques such as arterial spin labeling (about 2 seconds of non-invasively labeled blood) and dynamic susceptibility contrast (DSC) MRI (about 5 seconds of Gadolinium bolus), but appear to be exacerbated in CO2-CVR MRI (1–2 minutes of hypercapnic bolus) [33]. Therefore, the present pipeline also conducted analysis using voxel-wise temporal shift, such that each voxel not only gives a CVR value but also yields a temporal delay index. Some literatures have suggested that the delay index is also of potential diagnostic value in clinical conditions [32, 35].

Resting-state based CVR is a new method for CVR estimation. This method has the obvious advantage that it does not require the CO2-inhalation apparatus. According to the processing pipeline used in the present work, which is based on the work of Liu et al. [26, 34], it does not even require the CO2 recording. Therefore, the RS-CVR method is substantially easier to implement in research and even routine clinical settings. In fact, one can also conduct retrospective processing on the resting-state fMRI data previously collected, including those available in large multi-site studies, such as ADNI [43], ABCD [44], HCP [45], and UK Biobank [46]. The present processing pipeline uses filtered whole-brain BOLD time course as a surrogate of the actual CO2 recording in the regression model. One reason for this approach is that CO2 recording is not available in many resting-state fMRI studies and the requirement of a CO2 monitor and the associated sampling tubing adds to the complexity of the procedure. Another reason is that resting-state CO2 recording measured at the mouth may not accurately reflect the CO2 time course in the brain, due to the relatively small amplitude of CO2 fluctuations during rest in combination with the expected dispersion effect along its path through lung, pulmonary vein, heart, and feeding arteries [47, 48]. Of course, the drawback of not having a CO2 recording is that the estimated CVR is a relative measure, but does not provide an absolute value, e.g. %ΔBOLD signal per mmHg CO2. Thus the RS-CVR method is ideally suited for applications where the pathology is more focal or one can find a (relatively) healthy region, such as a contralateral normal hemisphere, as an internal reference standard. If absolute CVR value is desired in the resting-state method, one can record EtCO2 during the data acquisition. This can mitigate potential issues related to the use of internal reference model. The RS-CVR pipeline has been utilized to identify regional CVR abnormalities in human immunodeficiency virus-infected patients [49] and glaucoma patients [40].

Patient confidentiality, data protection, and security of data transfer are typical challenges associated with cloud-computing of medical imaging analysis. These issues have been

extensively discussed in the original report of the MRICloud platform [29]. Briefly, the uploaded imaging data are in ANALYZE format, which includes a raw image matrix and a header file that contains only the matrix dimension information. To ensure the security of the processing pipeline, SSH is used as the core in data transfer and signaling commands on the computational server. However, researchers should consider whether they need to include cloud processing in their consent process and ensure compliance with local data protection regulations before uploading data.

A few limitations of this work should be acknowledged. The CVR-MRICloud, similar to cloud-processing of other MRI modalities, lacks the flexibility of allowing user inputs or graphical user interface. This is by the design of the cloud platform in which the submitted data are not processed in real-time. Instead, the data are placed in a queue and will be processed sequentially by the computation server. Thus, it is not practical to have the user wait in front of the terminal to provide inputs. There are other CVR processing methods available for users with technical expertise, such as "SeeVR" which allows the users to further tune and optimize the regressors used in the GLM analysis [50]. Future version of the CVR-MRICloud will allow the uploading of user-defined ROIs and user-selection of the FWHM Gaussian kernel for spatial smoothing. As an additional limitation related to the CO2 processing algorithm, we observed in 4 cases an unanticipated phenomenon dubbed 'CO2 Switching'. Usually, the EtCO2 corresponds to the peaks of the CO2 trace representing the CO2 content in the exhaled air. However, when a subject's lung CO2 content responds too slowly to the hypercapnic gas challenge, the exhaled air temporarily has a CO2 content that is lower than the inhaled air, resulting the trough of the CO2 trace representing the EtCO2. As a result, the automatic CO2 processing algorithm incorrectly labels the subject's inhaled CO2 partial pressure as their EtCO2 curve. In these cases, manual correction is needed. Two examples of "CO2 Switching" are shown in Fig 13. We found that "CO2 Switching" is rare and only occurs when the subject has an unusually low baseline EtCO2 level (e.g. <25 mmHg). There is no known algorithm to detect the "CO2 switching" automatically. Therefore, users are encouraged to examine the CO2 traces for potential "CO2 switching" when unusually low baseline EtCO2 level is observed.

## Conclusion

In this work, we presented a cloud-based pipeline for the processing of CO2-inhalation and resting-state CVR data, referred to as CVR-MRICloud. This tool is fully automated, operating-

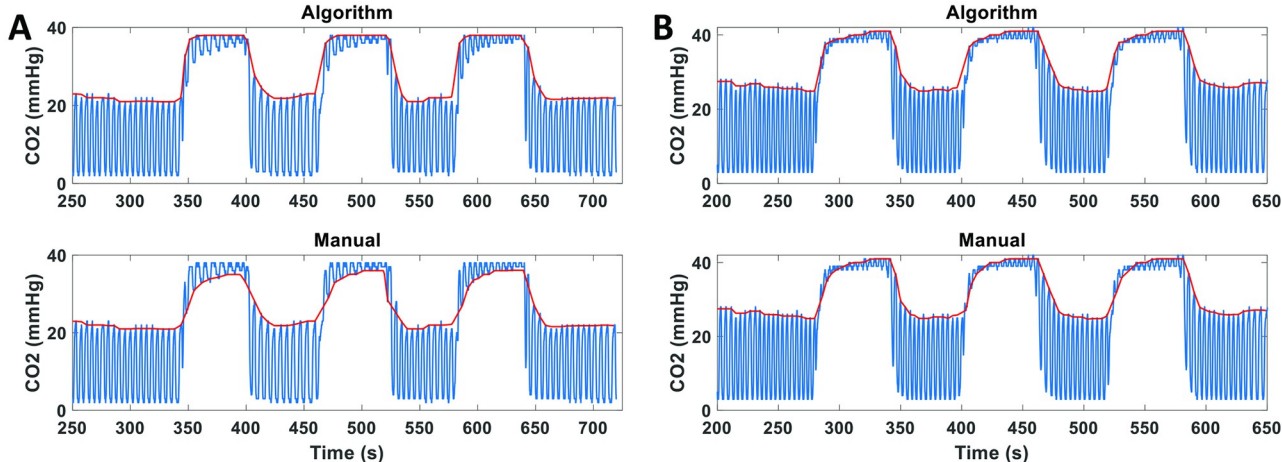

**Fig 13. Two examples of "CO2 Switching".** Top panels show the EtCO2 time courses obtained from the automatic CO2 processing algorithm. Bottom panels show the manual and correct EtCO2 time courses.

system independent, and has minimal requirements on the user's computer. CVR-MRICloud provides quantitative CVR and delay time results in regional values and voxel-wise maps. Applications of CVR-MRICloud in lifespan healthy subjects as well as in patients with brain pathologies were demonstrated. CVR-MRICloud has potential to be used as a data processing tool for a variety of basic science and clinical applications.

## Supporting information

**S1 Fig. EtCO2 curves and motion curves from two healthy subjects with high $R^2$ values.** The motion curve was in foot-head (FH) direction obtained using spm_realign.m function. (DOCX)

## Author Contributions

**Conceptualization:** Peiying Liu, Hanzhang Lu.

**Data curation:** Peiying Liu, Denise C. Park, Babu G. Welch, Jay J. Pillai, Argye E. Hillis, Susumu Mori, Hanzhang Lu.

**Formal analysis:** Peiying Liu, Zachary Baker.

**Funding acquisition:** Peiying Liu, Denise C. Park, Argye E. Hillis, Susumu Mori, Hanzhang Lu.

**Investigation:** Peiying Liu, Zachary Baker, Yang Li, Jiadi Xu, Marco Pinho, Hanzhang Lu.

**Methodology:** Peiying Liu, Zachary Baker, Yue Li, Yang Li, Jiadi Xu, Susumu Mori, Hanzhang Lu.

**Project administration:** Peiying Liu, Hanzhang Lu.

**Resources:** Peiying Liu, Yue Li, Denise C. Park, Babu G. Welch, Marco Pinho, Jay J. Pillai, Argye E. Hillis, Susumu Mori, Hanzhang Lu.

**Software:** Peiying Liu, Yue Li, Susumu Mori, Hanzhang Lu.

**Supervision:** Peiying Liu, Hanzhang Lu.

**Validation:** Peiying Liu.

**Visualization:** Peiying Liu, Marco Pinho, Hanzhang Lu.

**Writing – original draft:** Peiying Liu.

**Writing – review & editing:** Zachary Baker, Yue Li, Yang Li, Jiadi Xu, Denise C. Park, Babu G. Welch, Marco Pinho, Jay J. Pillai, Argye E. Hillis, Susumu Mori, Hanzhang Lu.

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
