## [Decision Letter · Decision Letter 0]

21 Jun 2022

PONE-D-22-11894CVR-MRICloud: an online processing tool for CO2-inhalation and resting-state cerebrovascular reactivity (CVR) MRI dataPLOS ONE

Dear Dr. Liu,

Thank you for submitting your manuscript to PLOS ONE. After careful consideration, we feel that it has merit but does not fully meet PLOS ONE’s publication criteria as it currently stands. Therefore, we invite you to submit a revised version of the manuscript that addresses the points raised during the review process.

We look forward to receiving your revised manuscript.

Kind regards,

Yen-Yu Ian Shih, Ph.D.

Academic Editor

PLOS ONE

Journal Requirements:

Additional Editor Comments:

Dear Peiying,

My apology for the delay in making a decision. As you can see, both reviewers are enthusiastic about this work and suggest the manuscript to be suitable for publication pending a minor revision. Please feel free to let me know if there’s anything I can help clarify further.

Best,

Ian

Reviewers' comments:

Reviewer's Responses to Questions

**Comments to the Author**

1. Is the manuscript technically sound, and do the data support the conclusions?

Reviewer #1: Yes

Reviewer #2: Yes

2. Has the statistical analysis been performed appropriately and rigorously? 

Reviewer #1: N/A

Reviewer #2: Yes

3. Have the authors made all data underlying the findings in their manuscript fully available?

Reviewer #1: No

Reviewer #2: Yes

4. Is the manuscript presented in an intelligible fashion and written in standard English?

Reviewer #1: Yes

Reviewer #2: Yes

5. Review Comments to the Author

Reviewer #1: In this manuscript Liu et al report their development and publication of a cloud-based tool for processing of CVR MRI data. The authors outline their rationale of expanding availability of CVR processing, both for a hypercapnic based and resting-state MRI paradigms, the implemented methodology and provide some validation in a healthy lifespan cohort of 203 datasets, three patient datasets and a healthy young subject. Liu et al report example outputs from the processing, results of the alignment and quantitative metrics amongst others. I am grateful to the editor and authors for the opportunity to comment on this interesting and useful work.

Major

I agree with the authors that there is considerable scope for broadening availability of CVR for clinical studies. However, there may be a risk of such a resource becoming a ‘black box’ for some prospective users, potentially leading to false inferences, and/or missing key details e.g. the reported ‘CO2 Switching’ phenomena. Further the authors make no mention of any evaluations of the software/user friendliness of the software by individuals with no experience of CVR which from the introduction appear to be the target audience.

The authors describe the benefits of cloud computing, which are significant, but not the challenges. Notably patient confidentiality, data protection and secure transfer/deletion are key. I believe that these should be considered in the Discussion, further detail on the technical setup and upload/download procedures of the server may also be appropriate for the Methods e.g. are uploads deleted after a set time? Do they only exist as long as a console is open?, How are results downloaded?, Who has access to/how secure is uploaded data? etc?

The authors present the toolbox as a way of measuring CO2- and RS-CVR – however no comparison between the two is given despite data being available in the same population. As a validation of the toolbox this would seem a useful validation i.e. to show that both approaches show similar associations, and I would encourage the authors to add such an analysis, along with more detailed discussion of the results in the Discussion.

Minor

The data availability statement is contradicted by the description of where data is to be found which indicates data will only be released case-by-case upon request after funding agency approval. Clearly there can be valid funding or ethics/consent restrictions which may limit how much data can be shared. However, if there is an exceptional reason it is not currently presented which should be corrected before publication.

Line 56 – The authors reference the Mild Stroke Study 3 (Clancy et al), however as a single centre study it does not seem to fit with their phrasing. The authors perhaps meant to refer to Investigate-SVDs 10.1016/j.cccb.2021.100020 which was multi-site and led by the same group?

Lines 68-69 – The authors draw an analogy between task/resting state fMRI and CO2-CVR/RS-CVR, however no citation is provided, I would suggest the authors either remove this sentence or add relevant references.

Lines 70-71 – The authors state most CVR studies have used in-house processing methods but do not supply a relevant reference, one such example would be 10.3389/fphys.2021.643468. The authors may also want to note that despite many CVR studies depending on in-house code few authors have published their code, creating challenges for reproducible and repeatable research.

Line 138 and 147 – What was the reason for choosing a window size of 10? And combining the results based on the higher of Algorithm I and II?

Line 169 – Delays of 0 to EtCO2 time course duration are allowed, and in line 175 a narrower search range of +/- 5 s is used. Are the authors confident this can handle the full range as pathology? Longer/altered delays have been reported in patient e.g. WMH in SVD patients etc.

Line 227 –Given the temporal filtering frequency in ref. 33 seem to come from healthy patients or moyamoya how transferable is the range used e.g. for stroke or MS?

Line 288-9 – Commenting on consistency of values with previous work may be better suited to the discussion than results.

Line 302-3 – Comments on reliability and reproducibility seem to be based on a single healthy subject, evaluations in other groups with larger sample size would be useful to determine how consistent this pattern is.

Line 337-9 – If no published data is available then it may be valuable to other groups to include some in the supplementary information to support this point.

Lines 207-211, Line 363 – Add reference. In general there were several areas with unsupported statements which may benefit from further additions, not least as it makes it more accessible for a wider readership.

Line 367 – internal reference models have significant limitations cf SUV models for PET, the drawbacks should be considered in my opinion. As although superficially the maps may look similar the lack of directly comparable values is a significant impediment.

Line 383 – Is there any error protection against the CO2 switching phenomena i.e. does MRICloud label it/throw an error?

Figure 9 – It would be useful to have the spread of the CVR as points instead of only the mean and CI.

Reviewer #2: In this work, the authors developed and describe a cloud-based cerebrovascular reactivity (CVR) processing pipeline. The authors comprehensively describe all of the processing steps, including extraction of end-tidal CO2 curves and its alignment with BOLD data, the calculation of a variety of CVR metrics from CO2-challenge and resting-state CVR data and a quality control metric. The authors also evaluate the performance of the processing pipeline from a population comprising a variety of ages, appropriate distribution of sex and MRI data from a wide range of imaging parameters. The manuscript is very clearly written and the work is valuable as it has the potential to broaden the uptake of the CVR technique by removing the data processing skill entry barrier, facilitates reproducible research and meticulously describes the logic behind each choice of processing step. Some minor questions and comments are listed below:

1) As the BOLD signal comprises a mixture of a variety of different hemodynamic and metabolic parameters, could the choice of MR imaging parameters affect the alignment of BOLD data to EtCO2, or the quality of CO2 and resting-state CVR maps (e.g. short TR, high flip angle increasing the sensitivity to in-flow effect)?

2) Could the authors clarify whether patient health information is automatically removed from image metadata upon upload to the cloud server or if the user is responsible for removing patient health information?

3) Could the authors describe what MRI image file formats are compatible with CVR-MRICloud?

4) If available, could the authors include a citation to the Smooth-Cross algorithm used for peak detection (algorithm 2, page 7, line 137) for the interested reader?

5) On page 18, line 370 the authors state that the CVR-MRICloud lacks the flexibility of allowing user inputs. Could the authors clarify whether this extends to the ability of the user to upload brain masks and the ability to modify the FWHM of the spatial-smoothing Gaussian kernel?

6) Do the authors plan to update the software, such as addition or modification of processing steps, as the CVR-mapping technique further develops? If so, will older versions still be available to users to ensure continuity of in-progress studies?

6. PLOS authors have the option to publish the peer review history of their article (what does this mean?). If published, this will include your full peer review and any attached files.

Reviewer #1: No

Reviewer #2: No

---

## [Author Response · Author response to Decision Letter 0]

28 Jul 2022

Response to Reviewer 1:

1. “I agree with the authors that there is considerable scope for broadening availability of CVR for clinical studies. However, there may be a risk of such a resource becoming a ‘black box’ for some prospective users, potentially leading to false inferences, and/or missing key details e.g. the reported ‘CO2 Switching’ phenomena. Further the authors make no mention of any evaluations of the software/user friendliness of the software by individuals with no experience of CVR which from the introduction appear to be the target audience.”

Thank you for this point. In this report, we have provided all details related to the CVR-MRICloud pipeline. The purpose of this report is exactly to de-mystify this tool and to make it not a “black box”. To the best of our knowledge, we have explained all known factors that could lead to false inferences. 

Regarding the use of the software by individuals with no experience of CVR, we would like to clarify that a substantial number of novice users have actually used our tool with no problems. We have added the following paragraph in the Discussion regarding the user-friendliness of the CVR-MRICloud:

“Minimal experience is required in order to use this tool. An inter-rater evaluation performed by 4 raters with no previous CVR experience revealed that the ICC of CVR values was >0.9959, suggesting that CVR-MRICloud is user-friendly and the results are not user-dependent [37]. The RS-CVR pipeline has also been utilized by researchers with no previous experience with CVR and yielded multiple publications [39-41].”

2. “The authors describe the benefits of cloud computing, which are significant, but not the challenges. Notably patient confidentiality, data protection and secure transfer/deletion are key. I believe that these should be considered in the Discussion, further detail on the technical setup and upload/download procedures of the server may also be appropriate for the Methods e.g. are uploads deleted after a set time? Do they only exist as long as a console is open?, How are results downloaded?, Who has access to/how secure is uploaded data? etc?”

Thank you. Indeed, patient confidentiality and data protection are important issues associated with human data processing over cloud. These issues have been addressed in the original report of the MRICloud platform (Mori et al, Computing in Science & Engineering. 2016;18(5):21-35). Note that the CVR-MRICloud is a new addition to the MRICloud toolbox and uses the same front-end framework as the other MRICloud tools in terms of data uploading/downloading. Briefly, the uploaded image file format is ANALYZE (.hdr and .img). This file format does not contain protected health information (PHI), thus it is compatible with Health Insurance Portability and Accountability Act (HIPAA) regulations. We have added this information in the revised manuscript:

“The required image file format is ANALYZE (.hdr and .img) which does not contain protected health information (PHI) and thus, is compatible with Health Insurance Portability and Accountability Act (HIPAA) regulations.”

Each submission of data to the server will create a job. The users can only access the results of the jobs their submitted. After downloading users can select to delete the data/result from the server. If the user do not delete the data/results, the data/result of the jobs will be automatically deleted from the server after 60 days. We have added the following sentences in the revised manuscript:

“The users can only access the outcome parameters from the data they had uploaded. The data and outcome parameters will be automatically deleted after 60 days from the upload if the users choose not to delete them manually.”

We have also added the following paragraph in Discussion:

“Patient confidentiality, data protection, and security of data transfer are typical challenges associated with cloud-computing of medical imaging analysis. These issues have been extensively discussed in the original report of the MRICloud platform [29]. Briefly, the uploaded imaging data are in ANALYZE format, which includes a raw image matrix and a header file that contains only the matrix dimension information. To ensure the security of the processing pipeline, SSH is used as the core in data transfer and signaling commands on the computational server.”

3. “The data availability statement is contradicted by the description of where data is to be found which indicates data will only be released case-by-case upon request after funding agency approval. Clearly there can be valid funding or ethics/consent restrictions which may limit how much data can be shared. However, if there is an exceptional reason it is not currently presented which should be corrected before publication.”

We have uploaded the anonymized data set necessary to replicate our study findings to a public repository. If this manuscript is accepted, the data will be available for downloading at https://doi.org/10.5061/dryad.wh70rxwqw.

4. “Line 56 – The authors reference the Mild Stroke Study 3 (Clancy et al), however as a single centre study it does not seem to fit with their phrasing. The authors perhaps meant to refer to Investigate-SVDs 10.1016/j.cccb.2021.100020 which was multi-site and led by the same group?”

We thank the reviewer for pointing this out. We have updated this sentence and reference accordingly:

“CVR as a candidate biomarker of vascular diseases has also been applied in multi-center settings, such as in the MarkVCID study [23] and the INVESTIGATE-SVDs study [24].”

5. “Lines 68-69 – The authors draw an analogy between task/resting state fMRI and CO2-CVR/RS-CVR, however no citation is provided, I would suggest the authors either remove this sentence or add relevant references.”

Per the reviewer’s suggestion, we have removed this analogy.

6. “Lines 70-71 – The authors state most CVR studies have used in-house processing methods but do not supply a relevant reference, one such example would be 10.3389/fphys.2021.643468. The authors may also want to note that despite many CVR studies depending on in-house code few authors have published their code, creating challenges for reproducible and repeatable research.”

We thank the reviewer for this point. We have added the reference and revised this sentence as below:

“CVR data processing is considered a niche skill and most studies of CVR have used in-house scripts or software [28], creating challenges for reproducible and repeatable research and for researchers and clinicians with less skills in software programming.”

7. “Line 138 and 147 – What was the reason for choosing a window size of 10? And combining the results based on the higher of Algorithm I and II?”

If the smoothing window is too small, it may not separate fluctuations during the breathing out period from separate breaths, whereas if the smoothing window is too large, it cannot detect the breathing peaks effectively during the transition periods between room air and CO2 inhalation. A window size of 10s (i.e., 2-3 breathing cycles in majority of subjects) is a good balance between the two issues. The final EtCO2 curve is obtained by selecting the higher value from Algorithms I and II at each point in the interpolated time series. As shown in Figure 6, the wrong values identified in each algorithm are always lower than the real values, because EtCO2 indicates the upper envelope of the CO2 trace. We have added the following sentences in the revised manuscript:

“The window size of 10s was chosen as a trade-off between the degree of smoothing and the sensitivity during the room air-CO2 transition periods.”

“Finally, the higher value from Algorithms I and II at each point in the interpolated time series was selected to form the EtCO2 curve (Fig. 3B), as the EtCO2 curve represent the top envelope of the CO2 trace.”

8. “Line 169 – Delays of 0 to EtCO2 time course duration are allowed, and in line 175 a narrower search range of +/- 5 s is used. Are the authors confident this can handle the full range as pathology? Longer/altered delays have been reported in patient e.g. WMH in SVD patients etc.”

We clarify that the EtCO2 delay was found in two steps. In the first step, we search through a broad range covering all possible shift values (0 to entire EtCO2 time course duration) with a courser step size of 1 second (see Figure 4a for example), and this can identify any physiological and pathological delay roughly. In the second step, based on the delay determined from the step 1, we search a narrow window (+/-5s) around the Step1-determined delay a with a finer step size of 100ms to achieve higher precision of the delay obtained (see Figure 4b for example). Therefore, we can effectively find the longer delays in pathological conditions. We have added the following sentence in the revised manuscript:

“This two-step procedure is expected to yield more reliable optimal shift value than using the first step alone.”

9. “Line 227 –Given the temporal filtering frequency in ref. 33 seem to come from healthy patients or moyamoya how transferable is the range used e.g. for stroke or MS?”

The temporal filtering frequency was determined from healthy subjects across the life span to be more generalized. In Ref.34 (i.e., Ref. 33 in the original submission) we have validated in a clinical cohort of Moyamoya patients. Moyamoya disease is a large artery stenosis in which the CVR abnormality varies in a large range. Therefore, we think the temporal filtering frequency determined in Ref. 34 can be transferable to most clinical applications. Our example image shown in Figure 12 demonstrated the application in stroke.

10. “Line 288-9 – Commenting on consistency of values with previous work may be better suited to the discussion than results.”

Per reviewer’s suggestion, we have moved this sentence to the Discussion section.

11. “Line 302-3 – Comments on reliability and reproducibility seem to be based on a single healthy subject, evaluations in other groups with larger sample size would be useful to determine how consistent this pattern is.”

The reliability and reproducibility of the resting-state CVR mapping technique has been reported with larger sample size in our previous studies (Refs 26 and 34). In this work, we are focusing on the processing tool, and therefore, we think an example of 7 repeated scans from the same subject is enough to demonstrate the reliability and reproducibility of our rs-CVR pipeline in CVR-MRICloud.

12. “Line 337-9 – If no published data is available then it may be valuable to other groups to include some in the supplementary information to support this point.”

Per reviewer’s suggestion, we have added the Supplemental Figure S1 to show the examples of subjects in whom the EtCO2 and motion vector (in foot-head direction) showed significant reverse-correlation:

“Supplemental Figure S1: EtCO2 curves and motion curves from two healthy subjects with high R2 values. The motion curve was in foot-head (FH) direction obtained using spm_realign.m function.”

13. “Lines 207-211, Line 363 – Add reference. In general there were several areas with unsupported statements which may benefit from further additions, not least as it makes it more accessible for a wider readership.”

We have added references accordingly.

14. “Line 367 – internal reference models have significant limitations of SUV models for PET, the drawbacks should be considered in my opinion. As although superficially the maps may look similar the lack of directly comparable values is a significant impediment.”

We agree. We have added the following sentences to discuss this issue and proposed potential solutions:

“If absolute CVR value is desired in the resting-state method, one can record EtCO2 during the data acquisition. This can mitigate potential issues related to the use of internal reference model.”

15. “Line 383 – Is there any error protection against the CO2 switching phenomena i.e. does MRICloud label it/throw an error?”

There is no known algorithm to detect the CO2 switching phenomena automatically. Therefore, we have reported this limitation and specified that the users should examine their CO2 recording and identify the cases where CO2 switching phenomena (as shown in Figure 13) is presented. We also noted that this CO2 switching phenomena is a rare situation. It happens only occasionally in subjects with baseline EtCO2 lower than 25 mmHg, whereas in majority of the subjects the baseline EtCO2 is between 35-45 mmHg. We have added the following sentences in the revised manuscript:

“We found that “CO2 Switching” is rare and only occurs when the subject has an unusually low baseline EtCO2 level (e.g. <25 mmHg). There is no known algorithm to detect the “CO2 switching” automatically. Therefore, users are encouraged to examine the CO2 traces for potential “CO2 switching” when unusually low baseline EtCO2 level is observed.”

16. “Figure 9 – It would be useful to have the spread of the CVR as points instead of only the mean and CI.”

We agree. We have added to the plot the subject-level CVR values to Figure 9.

 

Response to Reviewer 2:

1. “As the BOLD signal comprises a mixture of a variety of different hemodynamic and metabolic parameters, could the choice of MR imaging parameters affect the alignment of BOLD data to EtCO2, or the quality of CO2 and resting-state CVR maps (e.g. short TR, high flip angle increasing the sensitivity to in-flow effect)?”

Indeed, the underpinning of the CVR obtained may be dependent on the BOLD parameter used. For example, it could vary between oxygenation-weighted and flow-weighted, as the reviewer alluded to. In terms of their impact on the alignment, we feel that it will not be major. This is because the signal change due to CO2 response is a relatively slow response compared to brain activation induced BOLD signal change (where the neural activation takes less than 50 ms to occur). Thus, within the measurement error, the oxygenation-weighted and flow-weighted signal will have a very similar signal time-course. 

The optimal imaging parameter should be chosen to balance between SNR and purity of signal contrast, although this topic is beyond the scope of this paper.

2. “Could the authors clarify whether patient health information is automatically removed from image metadata upon upload to the cloud server or if the user is responsible for removing patient health information?”

We have clarified in the revised manuscript that, the required image file format of MRICloud is ANALYZE (.hdr and .img). This is because this file format does not contain protected health information (PHI), thus it is compatible with Health Insurance Portability and Accountability Act (HIPAA) regulations. We provide a DICOM to Analyze conversion tool on the MRICloud website which the users and download and use on their own computer. The users can also use their own software to generate the ANALYZE images of their data. We have added this information in the revised manuscript (please also see our response to R1.2):

“The required image file format is ANALYZE (.hdr and .img) which does not contain protected health information (PHI) and thus, is compatible with Health Insurance Portability and Accountability Act (HIPAA) regulations.”

3. “Could the authors describe what MRI image file formats are compatible with CVR-MRICloud?”

The required file format is ANALYZE. Please see our response to R1.2 and R2.2.

4. “If available, could the authors include a citation to the Smooth-Cross algorithm used for peak detection (algorithm 2, page 7, line 137) for the interested reader?”

The smooth-cross algorithm was developed by us during the course of this project. Thus there is no prior reference.

5. “On page 18, line 370 the authors state that the CVR-MRICloud lacks the flexibility of allowing user inputs. Could the authors clarify whether this extends to the ability of the user to upload brain masks and the ability to modify the FWHM of the spatial-smoothing Gaussian kernel?”

Thank you for this point. The “lack of flexibility” we mentioned was referring to the notion that the cloud tool is not ideal for “interactive” processing. For the pre-selected options such as uploading a brain mask or choosing FWHM, it is actually feasible. However, in the current implementation, we have not included these options. This is because we were trying to limit the number of user inputs to obtain a trade-off between tool flexibility and simplicity to users, especially novice users. We were concerned that, if the user types in a wrong input value by mistake, it will cause issues in the processing of the datasets. But we agree with the reviewer that allowing the user to upload brain masks and modify the smoothing kernel could be beneficial. So in the future we will update our implementation (as a separate version of the tool) to allow the user to upload user-defined brain masks and specify the data space from which the masks are defined, as well as select different FWHM of the smoothing kernel. We have added the following sentences in the revised manuscript:

“Future version of the CVR-MRICloud will allow the uploading of user-defined ROIs and user-selection of the FWHM Gaussian kernel for spatial smoothing.”

6. “Do the authors plan to update the software, such as addition or modification of processing steps, as the CVR-mapping technique further develops? If so, will older versions still be available to users to ensure continuity of in-progress studies?”

We thank the reviewer for pointing this out. We are updating the software with modifications and technical development. We are doing this in a version-controlled manner. We have added the following paragraph in the revised manuscript to clarify this:

“The deployment of CVR-MRICloud is conducted in a version‐controlled manner. When a new version is released, it will have a specific version number and all older versions are still available on the CVR‐MRICloud website.”

---

## [Decision Letter · Decision Letter 1]

24 Aug 2022

CVR-MRICloud: an online processing tool for CO2-inhalation and resting-state cerebrovascular reactivity (CVR) MRI data

PONE-D-22-11894R1

Dear Peiying,

We’re pleased to inform you that your manuscript has been judged scientifically suitable for publication and will be formally accepted for publication once it meets all outstanding technical requirements.

Kind regards,

Yen-Yu Ian Shih, Ph.D.

Academic Editor

PLOS ONE

Additional Editor Comments (optional):

Reviewer 1 left a note. Please kindly take a look. 

Reviewers' comments:

Reviewer's Responses to Questions

**Comments to the Author**

1. If the authors have adequately addressed your comments raised in a previous round of review and you feel that this manuscript is now acceptable for publication, you may indicate that here to bypass the “Comments to the Author” section, enter your conflict of interest statement in the “Confidential to Editor” section, and submit your "Accept" recommendation.

Reviewer #1: All comments have been addressed

Reviewer #2: All comments have been addressed

2. Is the manuscript technically sound, and do the data support the conclusions?

Reviewer #1: Yes

Reviewer #2: Yes

3. Has the statistical analysis been performed appropriately and rigorously? 

Reviewer #1: Yes

Reviewer #2: Yes

4. Have the authors made all data underlying the findings in their manuscript fully available?

Reviewer #1: Yes

Reviewer #2: Yes

5. Is the manuscript presented in an intelligible fashion and written in standard English?

Reviewer #1: Yes

Reviewer #2: Yes

6. Review Comments to the Author

Reviewer #1: I am grateful to the authors for their response which has addressed all the main substantive points.

The one caveat I would raise is in relation to point 2. While the authors have clearly considered the issues for their own legislative domain other regions have different standards, I think in particular of the General Data Protection Regulations (GPDR) standards which apply in most European countries. While previously data transfer to the USA was possible under the Privacy Shield this is unfortunately no longer the case: https://gdpr-info.eu/issues/third-countries. My (limited) understanding is at a minimum transfer of even anonymised data to a web server in the USA for processing would ideally be included in the participant consent form and ethics, and many institutions would have their own processes to approve transfer, which may include considering who else could hypothetically access the data e.g. server admins, law enforcement etc and on what basis.

Clearly any detailed discussion of these points is beyond the scope of this paper, and it would be impossible to account for local rules in all jurisdictions. However, the authors may wish to note somewhere that researchers should consider whether they need to include cloud processing in the USA in their consent process and ensure compliance with local data protection regulations before uploading data.

I congratulate the authors again on a valuable piece of work which I am sure will be extremely useful to the community going forward.

Reviewer #2: All questions and comments from the original review have been answered very well. Happy to recommend the manuscript for publication in its current form.

7. PLOS authors have the option to publish the peer review history of their article (what does this mean?). If published, this will include your full peer review and any attached files.

Reviewer #1: No

Reviewer #2: No

---

## [Editor Report · Acceptance letter]

19 Sep 2022

PONE-D-22-11894R1 

CVR-MRICloud: an online processing tool for CO2-inhalation and resting-state cerebrovascular reactivity (CVR) MRI data 

Dear Dr. Liu:

I'm pleased to inform you that your manuscript has been deemed suitable for publication in PLOS ONE. Congratulations! Your manuscript is now with our production department. 

Kind regards, 

on behalf of

Dr. Yen-Yu Ian Shih 

Academic Editor

PLOS ONE